# Analysis of COVID-19 on Diagnosis, Vaccine, Treatment, and Pathogenesis with Clinical Scenarios

**Daniel Tellez [1], Sujay Dayal [1], Phong Phan [1], Ajinkya Mawley [1], Kush Shah [1], Gabriel Consunji [1], Cindy Tellez [2], Kimberly Ruiz [2], Rutuja Sabnis [3], Surbi Dayal [4] and Vishwanath Venketaraman [1,***

[1] Basic Medical Sciences, College of Osteopathic Medicine of the Pacific, Western University of Health Sciences, Pomona, CA 91766, USA; daniel.tellez@westernu.edu (D.T.); sujay.dayal@westernu.edu (S.D.); phong.phan@westernu.edu (P.P.); ajinkya.mawley@westernu.edu (A.M.); kush.shah@westernu.edu (K.S.); gabriel.consunji@westernu.edu (G.C.)

[2] California State University, Northridge, CA 91330, USA; Cindy.tellez.204@my.csun.edu (C.T.); Kimberly.Ruiz.328@my.CSUN.edu (K.R.)

[3] University of California, Irvine, CA 92697, USA; rnsabnis@uci.edu

[4] Pitzer College, Claremont, CA 91711, USA; sudayal@students.pitzer.edu

**\*** Correspondence: vvenketaraman@westernu.edu

**Abstract:** As the world continues to suffer from an ever-growing number of confirmed cases of the SARS-CoV-2 novel coronavirus, researchers are at the forefront of developing the best plan to overcome this pandemic through analyzing the pathogenesis, prevention, and treatment options pertaining to the virus. In the midst of a pandemic, the main route for detection of the virus has been conducting antigen tests for rapid results, using qRT-PCR, and conducting more accurate molecular tests, using rRT-PCR, on samples from patients. Most common treatments for those infected with COVID-19 include Remdesivir, an antiviral, dexamethasone, a steroid, and rarely, monoclonal antibody treatments. Although these treatments exist and are used commonly in hospitals all around the globe, clinicians often challenge the efficacy and benefit of these remedies for the patient. Furthermore, targeted therapies largely focus on interfering with or reducing the binding of viral receptors and host cell receptors affected by the SARS-CoV-2 novel coronavirus. In addition to treatment, the most efficacious method of preventing the spread of COVID-19 is the development of multiple vaccines that have been distributed as well as the development of multiple vaccine candidates that are proving hopeful in preventing severe symptoms of the virus. The exaggerated immune response to the virus proves to be a worrying complication due to widespread inflammation and subsequent clinical sequela. The medical and scientific community as a whole will be expected to respond with the latest in technology and research, and further studies into the pathogenesis, clinical implications, identification, diagnosis, and treatment of COVID-19 will push society past this pandemic.

**Keywords:** SARS-COV-2; pathophysiology; treatment; COVID-19; prevention; diagnosis; clinical; case report

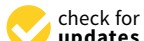



## 1. Introduction

The novel coronavirus known as SARS-CoV-2 has caused the deaths of over one million people worldwide and has infected over thirty-five million [1] Experts report that the virus has a 2.8% mortality rate among all infected. These statistics are alarming because the ease of transmission from the virus exponentially increases the number of lives lost each day. It is therefore crucial for researchers to keep up to date on all scientific and clinical information pertaining to COVID-19. Currently, there are hundreds of clinical and scientific trials all over the world attempting to gather information on the pathogenesis of the disease, prevention, and treatment. This review compiled the latest information on COVID-19 pathogenesis, immune response, diagnosis, and available treatments. We then relate this

information to specific clinical scenarios in an attempt to broaden our knowledge, confirm known findings, and provide a framework for clinicians and researchers attempting to better understand the disease.

## 2. Diagnosis

Any pandemic necessitates urgent detection and isolation of the infected individuals to prevent the transmission of pathogens. According to the WHO, patients infected with COVID-19 have had a wide range of symptoms that may appear 2–14 days after exposure to the virus. These symptoms can be non-specific such as fever or chills, cough, shortness of breath or difficulty breathing, fatigue, muscle or body aches, headache, sore throat, congestion or runny nose, nausea or vomiting, and diarrhea [2]. One more unique symptom is a new loss of taste or smell. This is not an exhaustive list of symptoms. With these symptoms in mind, which can mimic symptoms of the common flu, the recent worldwide pandemic, COVID-19, has intensely emphasized the key role of diagnostic technologies in the containment of infectious diseases. Sensitive, specific, and scalable diagnostic technologies help to identify cases early, aiding in prompt containment and treatment. As outlined below, there are various diagnostic methods to screen and identify patients with COVID-19.

One of the screening methods for identifying COVID-19 infection is a nasal and oropharyngeal swab procedure. The nasal and oropharyngeal swab procedure was approved by the World Health Organization (WHO) and operates by detecting COVID-19 RNA. However, this procedure acts more as a screening test and not a diagnostic test. Nasopharyngeal and oropharyngeal swabs require careful collection to reduce the false-negative rate. The data show that about 30% of swabs produce false-negative results from clinically symptomatic patients [3]. Oropharynx, nasopharynx, sputum, and bronchial fluids can be used to diagnose COVID-19 infection, although oropharynx and nasopharynx swabs are simpler to collect. The results of testing nasopharyngeal swab specimens to detect SARS-CoV-2 may vary with repeated sampling in individual patients; therefore, one study evaluated viral detection in matched samples over time. Wyllie and colleagues from the Yale School of Public Health noted that the level of SARS-CoV-2 RNA decreased after symptom onset in both saliva specimens and nasopharyngeal swab specimens. In three instances, it was shown that a negative nasopharyngeal swab specimen was followed by a positive swab at the next collection of a specimen; this phenomenon occurred only once with the saliva specimens. During the clinical course of the study, it was observed that there was less variation in levels of SARS-CoV-2 RNA in the saliva specimens (standard deviation, 0.98 virus RNA copies per milliliter; 95% credible interval, 0.08 to 1.98) than in the nasopharyngeal swab specimens (standard deviation, 2.01 virus RNA copies per milliliter; 95% credible interval, 1.29 to 2.70) [4]. This suggests that saliva swabs specimens may offer a more definitive diagnosis of COVID-19 than nasopharyngeal swab specimens, due to less variation.

COVID-19 Ag Respi-Strip, a dipstick immunochromatographic test, aims to identify SARS-CoV-2 antigens in nasopharyngeal samples within fifteen minutes [5]. Generally, performing RT-qPCR requires high-tech machinery and skilled lab members comfortable with molecular techniques. The rapid antigen detection test has several benefits, such as ease of conducting, rapid turnaround time, low finance, and lack of the necessity of special equipment when compared to molecular testing [5]. However, the rapid antigen detection test suffers from poor sensitivity based on the research data [5]. Antigen tests are generally quicker to diagnose the coronavirus infection compared to molecular tests, but antigen tests have a higher possibility of missing this active infection.

Furthermore, many companies perform molecular tests to detect the virus's genetic material in a sample collected from the patient. The optimal standard for COVID-19 diagnostics is real-time reverse transcriptase polymerase chain reaction (rRT-PCR)-based assays [6]. The rRT-PCR testing is limited by the transportation of samples to the labs, largely due to the demand [6]. Research has shown that CT scans and serological tests have

been less accurate than the molecular tests in diagnosing COVID-19, because the molecular tests can target and identify the specific antigen [6].

We can identify and follow up COVID-19 patients using portable chest X-ray machines and CT scans. Portable X-ray machines minimize the risk of additional infection by eliminating the risks related to patient transport to the CT suite and the inadequacies in CT room decontamination. However, because information provided by X-ray is limited, CT scans are more commonly utilized. A CT scan of a COVID-19 patient will depict irregular, hazy, reticular ground glass opacities, helping scientists differentiate between COVID-19 and bacterial pneumonia [7].

There are numerous established diagnostic tests that have enabled scientists to diagnose COVID-19. The rRT-PCR-based laboratory method is a gold standard for testing because it is more accurate than the other methods [6]. A rapid screening method such as nasal and oropharyngeal swabs is available, but this is more of a screening test and not a diagnostic test. Although molecular diagnostic tests detect the genetic material of the virus, they are commonly used for diagnosing COVID-19. However, there is not a test that can be used to detect the COVID-19 virus with certainty 100% of the time. There is an urgent need to develop point-of-care (POC) and multiplex assays to be rapidly implemented due to the urgent clinical and public health needs to drive an unprecedented global effort to increase COVID-19 testing capacity.

### 3. Vaccine

The COVID-19 vaccine, distributed primarily between Pfizer and Moderna in the United States, offers a potential solution to this ravishing disease via herd immunity. The vaccine is administered in two doses, often 20–30 days apart, with the second dose serving more as a booster. While most vaccines administered are live attenuated, killed, subunit, or toxoid, this vaccine is unique in that it is the mRNA of the virus which is being introduced to the body rather than the virus itself. Pfizer's vaccine is called BNT162b2 and is a lipid nanoparticle-formulated mRNA vaccine that encodes the spike protein [8]. Moderna's vaccine has the same principles as that of Pfizer, and is called mRNA-1273 (Baden, 2020). Side effects are mild, ranging from pain at the injection site to fatigue and headaches (Polack, 2020). Individuals with severe allergic reactions are advised not to receive the vaccine (Mahase, 2020). In terms of effectiveness, the Pfizer vaccine boasts an astonishing 95% conferred immunity after individuals have received both doses (Polack, 2020). Moderna also displays high percentages: 94% in individuals receiving both doses [9]. The two big downfalls to the vaccine are the supply and time, two elements that hopefully can be fixed in an expedient manner.

Additionally, there are several other COVID-19 vaccine candidates that are currently in Phase I–III that could serve as viable candidates for treatment. There are four subcategories of these vaccines that differ from the mRNA formulation, which include whole virus, protein subunit, nucleic acid, and viral vector vaccines. Each formulation protects patients in a similar manner as Pfizer and Moderna, although induces immunity by different mechanisms.

The Bacillus Calmette–Guerin (BCG) vaccine is one such candidate that has been recently studied as a possible candidate for a COVID-19 vaccine. The vaccine was developed between 1908 and 1921 and is FDA-approved to treat bladder cancer and as a vaccine for patients at risk of contracting TB. However, in a retrospective observational study published on 19 November 2020, Rivas et al. report the BCG vaccination history which showed that 29.6% of 6201 candidates had a much lower seroprevalence of anti-SARS-CoV-2 IgG, as well as markedly decreased self-reported incidents of clinical symptoms caused by COVID-19. [10] However, it is important to note that a BCG vaccine will not be more effective than a specific vaccine for COVID-19; Dr. Mshe Arditi, co-senior author of the study, is studying the vaccine because it has been known to have a general protective effect against a range of bacterial and viral diseases other than TB, including neonatal sepsis and respiratory infections [11] Therefore, a larger number of randomized clinical trials

have been launched to study the potential protective effects of BCG vaccination against COVID-19.

SBC-2019 is an example of a protein subunit vaccine candidate that can serve as a COVID-19 vaccine candidate. This vaccine contains a stabilized trimeric form of the spike protein combined with two different adjuvants (S-Trimer). This vaccine is being developed in Australia, in which Phase I trials of two age groups from 18 to 54 years old, and older adults aged 55–75, were given either a placebo or vaccine, with two doses given to the participants spaced twenty-one days apart. With the 148 participants who remained in the study, the vaccinated was generally well-tolerated, with high titers and seroconversion rates of neutralizing antibodies of the spike protein in both the younger and older adult trial groups [12] As of now, SBC-2019 with the two best adjuvant candidates combined with the vaccine base are to be taken into the Phase II/III trials, and final selection will be determined based on manufacturing considerations.

INO-4800 is an example of a nucleic acid-based vaccine that also shows promise as a vaccine against COVID-19. This DNA vaccine encodes the SARS-CoV-2 S-protein and, similar to Moderna and Pfizer, requires two doses to stimulate a robust humoral and immune response. Between 6 April 2020 and 23 April 2020, forty participants were enrolled in a trial in which thirty-eight of these patients completed both dosages. Most participants of this group had an increase in serum IgG binding titers to S1 + S2 spike protein when compared to their pre-vaccination point, with an overall increased number of antibodies binding to S-protein [13]. No serious adverse events were reported in this trial. However, this Phase I trial only used a modest sample size and involved healthy volunteers that ranged from 18 to 50 years old and is currently following these Phase I participants for 12 months to evaluate the long-term safety of the vaccine and durability of the immune response. As of publication of the study on 20 December 2020, the efficacy of this vaccine is still being planned for additional trials. As of now, it is in Phase II/III of testing, but studies have currently been put on hold [14]

GRAd-COV2 is one such candidate that utilizes a novel incompetent simian adenovirus strain. The GRAd-COV2 vaccine study is being run in Rome, Italy, at the Lazzaro Spallanzani National Institute for Infectious Diseases, in which the vaccine is currently in Phase II/III clinical trials in healthy, elderly subjects aged 65–85 years old. Similar to the vaccine candidates developed by Johnson & Johnson or AstraZeneca, GRAd-COV2 is based on a replication defective adenovirus vector that encodes the full-length coronavirus spike protein. However, unlike the two vaccine candidates that use a human adenoviral vector and a chimpanzee one, respectively, the GRAd-COV2 vaccine is derived from gorillas. This vector is purported to potentially serve as a strong immune response due to its low seroprevalence in humans and great efficacy compared to other simian and human adenovirus vectors in clinical trials. As of 19 December 2020, the GRAd-COV2 vaccine is well-tolerated and induced a clear immune response in 45 healthy volunteers aged 18–55 [15] At this point, however, there is no prediction as to when the vaccine will be ready to be delivered, specifically in the European Union.

### 4. Pathogenesis

The pathogenesis of SARS-CoV-2, which is responsible for COVID-19, is similar to that of SARS-CoV, which was responsible for the outbreak of severe acute respiratory syndrome in Asia in 2003. Both viruses belong to the coronavirus group, and their pathogenesis involves the attachment of the virus to the cell surface and subsequent endocytosis [16]. Once within the cell, the viral RNAs are released, which then enter the nucleus and use the host machinery to translate into proteins as well as to replicate [16]. On the surface of coronavirus are spikes, which are glycoproteins that determine to which cells the viruses can attach. The spikes have two components: S1 mediates the binding to host cell receptors, whereas S2 mediates the fusion of the membranes [16]. Many studies have identified ACE2 as a functional receptor for the coronavirus, which leads to the hypothesis that ACE-inhibitor, which is usually used as antihypertensive medication, might have a

therapeutic role in preventing the cellular invasion of coronavirus and subsequent clinical manifestations [17,18]. ACE2 receptors are expressed in many tissues, predominantly in the lung epithelial cells, which explains the major pulmonary symptoms seen in patients infected with the viruses [19] (Figure 1). Following the binding to ACE2 receptors, the viral spikes undergo cleavage and conformational changes, which allow viral RNA to enter the cells [20]. ACE2 receptors are located in high concentrations in the alveoli, which explains why the early signs of pulmonary cell infiltration and destruction are detected in the distal airway [21] On CT scans, patients can present with ground glass opacification [21].

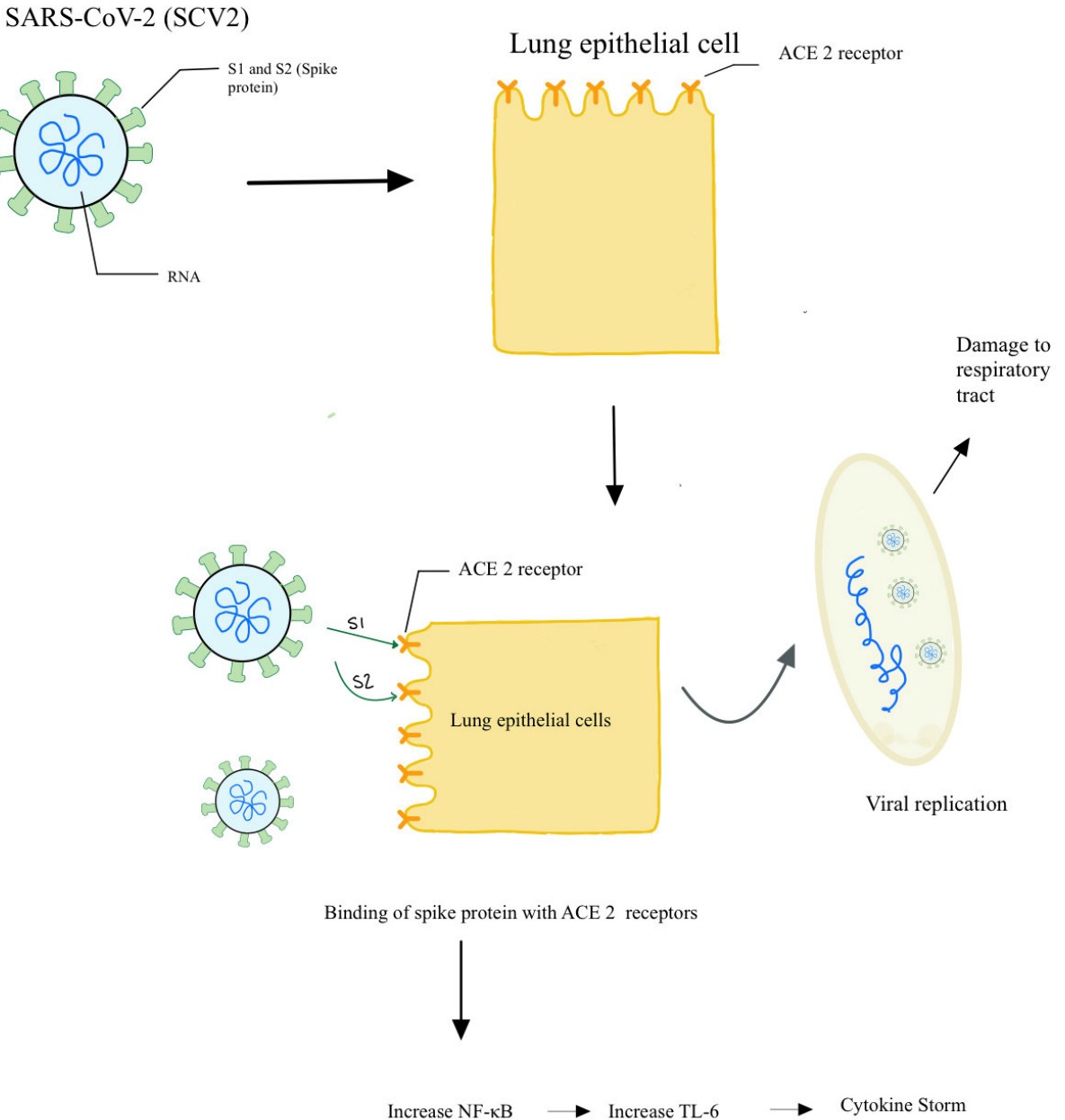

**Figure 1.** Schematic model explaining the pathogenesis in COVID-19.

Besides ACE2 receptors, it has been hypothesized that SARS-CoV also uses other types of receptors, such as DC-SIGN, that are found on dendritic cells and macrophages [22]. This gives rise to the theory that SARS-CoV-2 can also enter antigen-presenting cells

directly, which would provide one explanation of how APCs are able to acquire and present SARS-CoV-2 antigens on their major histocompatibility complexes. However, this theory still needs more research and verification. A more traditional explanation is that APC phagocytose virally infects epithelial cells, breaks down those cells, and then acquires and presents the viral antigens (Figure 1). These APCs then migrate to the lymph nodes and present the viral antigens to other immune cells, starting the inflammatory reactions that eventually lead to systemic manifestations of the disease, which include fever [23]

## 5. Treatment

Given the mass traction and deleterious effects of COVID-19, it should be of no surprise the massive efforts and strides made in the scientific community to find a treatment for this virus. The biggest downfall in treatment regimens lies in the fact that viruses cannot be merely treated by antibiotics. An example would be influenza, because the standard protocol of treatment is adequate hydration and rest. One of the main tasks in constructing an effective treatment for COVID-19 is to reduce the binding of the viral receptor (located on protein S) and the host cell receptor (located on ACE2 or DPP4) [24]. Here, we will be discussing four popular treatments (antivirals, dexamethasone, and monoclonal antibodies, apilimod), their mechanism of action, strengths and weaknesses, and any clinical trials conducted.

Initially constructed to combat Ebola, antiviral Remdesivir has made headlines for being a potential treatment for other RNA viruses such as COVID-19. Remdesivir is an adenosine analog that interferes with viral RNA synthesis by integrating itself into the viral RNA chain and causing premature termination [25]. It is a prodrug, meaning that it metabolizes into an active alanine metabolite once passively diffused inside the cell, and it is this metabolite that can be used by the viral RdRp enzyme as a substrate [26]. Specifically, it is the 1'CN substituent in the alanine metabolite that interacts with the viral RdRp enzyme to halt synthesis [26]. A study involving 61 COVID-19 patients illustrated improvement in 36 out of 53 patients (68%) who were administered Remdesivir [27]. However, one of the downfalls with Remdesivir as a treatment is that there are conflicting pieces of evidence questioning its efficacy. For example, a study involving 79 COVID-19 patients showed no statistically significant evidence that Remdesivir reduces the time it takes for an improvement of one's health [25]. Additionally, comprehensive knowledge of the drug itself is unknown, such as its interactions with other drugs and pharmacokinetics/pharmacodynamics. More research is currently being conducted to evaluate the use of Remdesivir in combination with immunosuppressants to combat the disease, because Remdesivir alone cannot reduce the cytokine storm seen in COVID-19 [28]. Remdesivir and other antivirals need to be better elucidated regarding the mechanism of action and effect on COVID-19 patients before they become a mainstay treatment during the pandemic.

Known as a lifesaving drug in the context of autoimmune and inflammatory diseases, dexamethasone has been recommended by many as a potential treatment for COVID-19 patients. Dexamethasone is a corticosteroid that has anti-inflammatory effects similar to those of cortisol, because they both inhibit the release of chemokines by immune cells. The use of dexamethasone may reduce lung inflammation and decrease the severity of acute respiratory distress syndrome (ARDS) [29]. In addition, some postulate that the anti-inflammatory property of dexamethasone may protect the lungs by combating the cytokine storm associated with COVID-19 [20]. A study involving 2104 patients hospitalized with COVID-19 discovered that dexamethasone did indeed lower the 28-day mortality for patients receiving oxygen or ventilation but did not have the same impact on those without respiratory support (Dexamethasone in Hospitalized Patients . . . , 2020). One of the biggest downfalls with dexamethasone is that despite inhibiting the production of damaging cytokines, it will also limit the beneficial function of T cells and B cells. The inhibition of B cells from making antibodies results in the long-term presence of increased plasma viral load. Furthermore, dexamethasone also prevents macrophages from performing their function, because macrophages are blocked from clearing secondary, nosocomial,

infections [30]. Another problem with dexamethasone lies in the plethora of side effects entailed with corticosteroids, such as Cushing syndrome, oral thrush, striae, insulin resistance, bone suppression resulting in osteoporosis, hypertension, glaucoma, etc. Therefore, for short-term relief, dexamethasone appears to be advantageous for COVID-19 patients; however, this steroid may be equally as deleterious in the long term due to the persistence of the virus along with the prevention of the production of antibodies.

The next popular treatment method analyzed is monoclonal antibodies (mab). Monoclonal antibodies found either in the blood of infected individuals or manufactured in the laboratory have been shown to reduce the epitopic region (spike protein) in COVID-19, reducing viral entry and replication, thus resulting in a reduction in disease severity [24]. It should be noted that this form of medication is not novel, because treatments for many diseases such as Middle East Respiratory Syndrome and Ebola have utilized such an approach. There have been many documented monoclonal antibodies for COVID-19 working on specific interactions between the virus and host cell, such as mab CR3014 decreasing the binding between the S1 domain of the virus and ACE2 receptor on the host cell [31]. One of the problems with mab therapy is that monotherapy is ineffective; one experiment illustrated that mab CR3022 alone did not show neutralization, but a mixture of CR3022 and mab CR3014 showed neutralization [31]. One can imagine the time and cost to construct a cocktail of mabs for COVID-19, a major drawback for such treatment modality. However, clinical trials such as mab LYCOV555 (recovered from a COVID-19 patient) insertion in high-risk individuals residing or working at nursing facilities by scientists at AbCellera are now enrolling across the world (Clinical trials of monoclonal antibodies . . . , 2020). A combination of more trials (data) and the comprehension of host interaction mechanisms and viral kinetics can bolster mab therapy and lead to effective and rapid response.

Tocilizumab is another potential monoclonal antibody treatment that can serve to treat COVID-19-caused pneumonia (COVID-19 PNA). Tocilizumab is a monoclonal antibody against interleukin-6 receptor alpha, used primarily to treat inflammatory diseases. Severe COVID-19 PNA patients were observed to have better outcomes observed from several case reports. With tocilizumab, patients were observed to have rapid reduction in fever, reduced use of oxygen support and mechanical ventilation, and reduction in pathologic lung manifestations. In a total of 479 patients in the study, the 295 patients that received tocilizumab received less glucocorticoid in the management of COVID-19 PNA [32]. In REMAP-CAP and RECOVERY trials of tocilizumab, a mortality benefit was reported in selected populations. REMAP-CAP enrolled a narrowly defined population of critically ill patients requiring respiratory support who were admitted to an ICU and randomized to receive open-label tocilizumab ($n$ = 353) or usual care ($n$ = 402). Corticosteroids were given to 92.7% and 93.9% of the patients in the tocilizumab and usual care arms, respectively. Compared to usual care, tocilizumab use reduced both in-hospital mortality (28% of the tocilizumab recipients vs. 36% of the usual care recipients died) and time to hospital discharge (HR 1.41; 95% credible interval [CrI], 1.18–1.70) and increased the number of organ support-free days (10 days in the tocilizumab arm vs. 0 days in the usual care arm; OR 1.64; 95% CrI, 1.25–2.14). (U.S Department of Health . . . ., 2021)

The RECOVERY trial enrolled hospitalized patients with COVID-19 into an open-label, platform trial of several treatment options. A subset of participants with hypoxemia (i.e., SpO2 < 92% or need for supplemental oxygen) and CRP level $\geq$ 75 mg/L were offered enrollment into a second randomization (1:1) to tocilizumab (8 mg/kg once, with possible second dose) versus usual care. Across the tocilizumab arm ($n$ = 2022) and the usual care arm ($n$ = 2094), the median duration of hospitalization was 2 days, and 82% of the participants were receiving concomitant corticosteroids. At baseline, 45% of the participants were on conventional oxygen, 41% on HFNC or NIV, and 14% on IMV. The study reported that tocilizumab reduced all-cause mortality over 28 days (29% of tocilizumab recipients vs. 33% of usual care recipients died by day 28; RR 0.86; 95% CI, 0.77–0.96), as well as the median time to being discharged alive (20 days for the tocilizumab recipients vs. >28 days

for the usual care recipients). The study has not yet been published in a peer-reviewed journal. (U.S. Department of Health . . . . 2021)

Apilimod is a chemotherapeutic agent (specifically, a PIKfyve kinase inhibitor), and when paired with cysteine, protease inhibitors, or vacuolin, has shown potential for reducing the impacts of COVID-19 [33]. The drug targets both viral entry and replication in human pneumocyte-like cells derived from stem cells, as exemplified by the studies on lung tissue showing percentages as high as an 85% reduction in the virus [33]. Specifically, it is the trafficking interaction between the lysosomes, endosome, and trans-Golgi network that the drug is blocking, resulting in swollen vesicles barring viral entry [34]. Side effects for the drug are inconclusive, ranging from non-severe headaches to nausea (as expected from chemotherapeutic agents), to severe suppression of the immune system, which can be counterproductive in treating COVID-19 [34,35]. The biggest downfall to the drug is the lack of clinical trials. There is currently a Phase II trial organized by the NIH consisting of 142 participants receiving either a placebo or apilimod, but the results have not been tabulated. Regardless, apilimod is a drug that warrants additional research and trials, because there is no miracle cure for the disease.

In addition to the aforementioned treatments, as of March 31, 2021, *The BMJ* has published several new guidelines regarding new approaches for the management of severely and non-severely diseased patients. One of these promising new treatment options is JAK inhibitors, which show potential for reducing mortality and time of patient ventilation. Another drug is colchicine, which has been shown to reduce mortality and risk of ventilation in patients with non-severe disease. One therapeutic which has been found to be ineffective was azithromycin, which has not shown any important impact on hospitalized patient outcomes. Likewise, another set of drugs which were studied are ivermectin with doxycycline, for which therapeutic effects remain uncertain for both severe and non-severe diseased patients (BMJ 2021;372:n858).

## 6. Immune Responses

Plasma levels of proinflammatory cytokines have been shown to be increased in patients infected with coronavirus. These include IL-6, IL-10, and TNF-alpha [36]. Furthermore, IL-6 levels have been correlated with the severity of the disease. CD4 and CD8 T cells are activated. CD8 T cells are also known as killer T cells and are responsible for killing virally infected cells, whereas CD4 T cells are also known as helper T cells and help mediate the inflammation and antibody development. Studies have shown that T cells are exhausted during infection, which could have contributed to the progression of the disease [37]. GM-CSF has been reported to also increase during coronavirus infection, which can enhance T cell function; however, this cytokine can cause tissue damage if overexpressed [38]. These immunological responses were primarily obtained from studies in adults infected with the virus. Another study showed that the infected epithelial cells also produce IL-8, a strong attractant for neutrophils, which explains the large number of inflammatory cells observed in patients with severe disease, apart from CD4 and CD8 T cells [39]. In general, neutrophils and T cells help contain the infection; however, in excess, they can also contribute to the lung damage seen in coronavirus-infected patients (Figure 2). Therefore, the lung damage seen in these patients is contributed not only by the cellular invasion by the viruses themselves, but also by the milieu of released cytokines and responding immune cells [40].

Therefore, cytokine imbalance resulting from increased levels of pro-inflammatory cytokines and decreased levels of anti-inflammatory cytokines (IL-10 and TGF-β) will result in inflammation, tissue damage, and immune exhaustion of T and B cells, leading to progression of the disease. Conversely, cytokine balance will result in effective host immune response and recovery from SARS-CoV-2 infection (Figure 2).

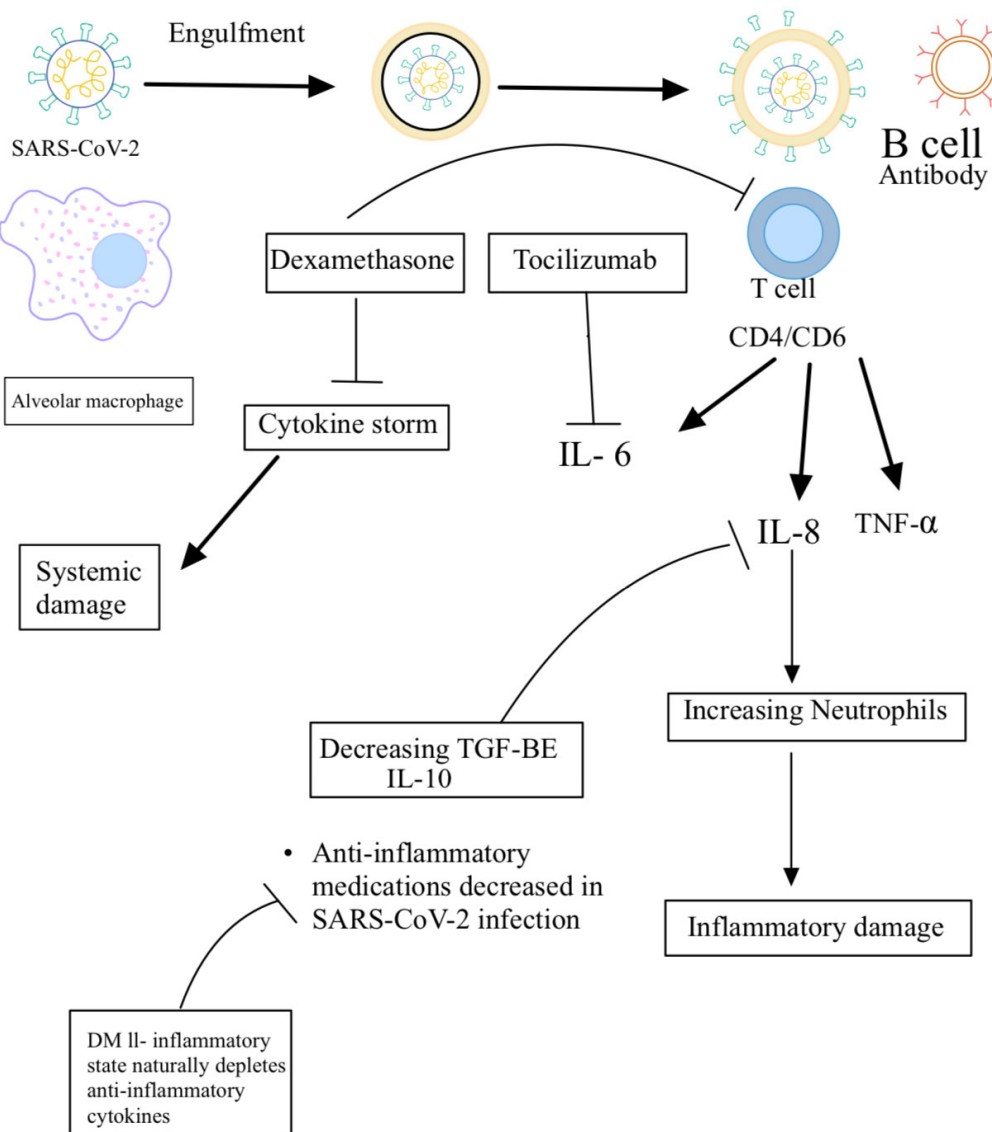

**Figure 2.** Underlying host immune responses against SARS-CoV-2 infection.

## 7. Clinical Case Scenario

Relating COVID-19 scientific evidence with clinical scenarios is important because it allows researchers to understand and further research the implications which COVID-19 has on the human body.

### 7.1. Patient 1

Patient 1 was 19-year-old male with no pre-existing health conditions; it was estimated that his infection began one week prior to experiencing any symptoms. His diagnosis was confirmed via an RT-PCR method performed about 1 week after the onset of symptoms. According to a recent review, 80% of COVID-19 cases are mild with cold-like symptoms and mild pneumonia [41]. Our patient fell into this category, because his presentation included cough, fevers, and mild temperature, clear phlegm, and body aches. The patient, however, also presented rare symptoms which included "feeling dehydrated" and "loss of smell" as well as "severe back pain". Normally, it has been agreed upon that SARS-CoV-2 spreads via droplets and human-to-human transmission [42]. However, in the case of our

patient, he reported attending social gatherings and interacting with close family with no consequential infections during the proposed 2–3-week active infection period. As our report proposes, we believe that increased IL-10 anti-inflammatory cytokine levels in this individual with a likely decreased immune exhaustion due to his age and health may have been the cause for reduced recovery time and spread of the SARS-CoV-2 virus. We believe this opens up new research realms, because patients with stronger immune systems may be able to fully contain the virus and not be infectious to other humans; our patient was around 12 different people with no PPE, and not one returned a positive RT-PCR COVID-19 test post-exposure. Before resuming normal activities, this patient was required to quarantine at home for at least 10 days to manage the infection until receiving a negative test and no symptoms were observed [41]. As was expected, due to our patient being a healthy young male, he was placed in one of the least morbid brackets and expected to make a full recovery. Symptoms cleared 7 days after receiving a positive RT-PCR test, and negative results were observed 3 days after. It was then expected that patient 1 would make a full recovery with a very low likelihood of reinfection [43], however it is poorly understood whether sequelae from the diseases will be experienced. Some sequelae that have been reported at a very low rate are permanent lung damage, MI, stroke, and even pulmonary embolisms [43]. It is to be mentioned that the severity of sequela and reinfection are also highly based on initial morbid factors before diagnosis.

### 7.2. Patient 2

Patient 2 was a 52-year-old diabetic, obese, hypertensive male who contracted COVID-19 after denial to follow social distancing and mask-wearing preventative measures. According to a study written by the WHO, this patient would classify under the minimal common outcome measure as "Hospitalized with Moderate disease" [44]. This measure of outcome attributes morbidity factors such as viral burden, age, symptoms, biomarkers, hospital stay length, and other aspects to categorize patients into brackets based on disease severity. It is thought that patient 2 obtained exposure during a visit from a family member, and 2 days afterwards experienced the main COVID-19 symptoms of cough, fever, myalgia, and headache [45]. He was diagnosed using RT-PCR testing and was confirmed positive two times over the course of his hospitalization. Several days afterwards, he reported that he was experiencing severe back pain located in the kidney region. It is worth noting that this symptom was prevalent in all family members exposed to him. Patient 2 received oxygenation due to mild ARDS and supportive therapy upon admittance to the ER, the standard protocol for such infection [41] Upon admittance, our patient was continued on oxygen therapy for three days, and also received dexamethasone with Remdesivir because this was the promising therapeutic at the time. We propose that due to his pre-existing diabetic condition, which is known to cause cytokine imbalance, decreased anti-inflammatory cytokines (IL-10 and TGF-B) likely contributed to an altered pathogenesis as compared to patient 1. We believe that this could have caused such an imbalance that the resulting clinical presentation was more severe. After hospitalization, the patient recovered after 7 days of hospitalization and reported positive cases of all people in his household. We hypothesize that in contrast to patient 1, this patient with DMII and a cytokine disbalance likely had an increased potential to spread the virus, which resulted in infections of those in his household.

## 8. Discussion

The COVID-19 infection shares many characteristics of its pathogenesis with the SARS-CoV pandemic in Asia in 2003. The virus enters the cell by using its characteristic spikes to bind to host cell receptors and fuse with its cell membrane to eventually infect the cell and replicate it. This replication and spread throughout the body lead to clinical manifestations of the COVID-19 disease. Although other cell-surface receptors have been identified, one of particular interest is the ACE2 receptors which are found in abundance in lung alveoli, explaining the pulmonary pathology of the virus. The primary immune response

to the SARS-CoV-2 virus is the phagocytosis of virally infected epithelial cells by antigen-presenting cells (APCs). This leads to the presentation of viral antigens to other immune cells, resulting in a systemic immune response. This immune response is largely composed of neutrophils and T cells which help to fight the infection; however, in excess amounts, this can lead to system end organ damage such as lung pathology. Other important cytokine mediators related to COVID-19 include IL-6, IL-8, IL-10, and TNF-alpha.

Once infected, there is a large variance in how severely COVID-19 affects patients, and there are no definite risk factors that can predict outcomes. It is well documented that age and pre-morbidities increase the risk of adverse outcomes, however there are many exceptions to the rule. Although most patients do make a full recovery, the future complications secondary to COVID-19 infection remain unknown and are a topic of active research as we learn more about this virus.

Treating COVID-19 has been the center point of attention for scientists and healthcare professionals since its early onset. Viral infections cannot be treated as simply as bacterial infections; therefore, finding a reliable and efficacious treatment for COVID-19 has proven to be difficult. The SARS-CoV-2 pandemic has required the use of rapid methods for its identification, diagnosis, and treatment. Initial screening tests have utilized nasopharyngeal and oropharyngeal swab collections for further analyses by using RT-PCR for antigen tests. Real-time reverse transcriptase PCR activity (rRT-PCR), a quantitative molecular test, has been one of the most successful methods in culturing and isolating this novel virus; it has been used as a standard approach in the medical community to diagnose COVID-19. The molecular test, rRT-PCR, not only detects the virus, but can quantify the amount of RNA, known as the viral load. Other diagnosing methods include the use of CT scans, X-rays, and serological tests, which have not been as successful as rRT-PCR. However, even though rRT-PCR has been fairly successful, there is a need to develop further multi-dimensional assays on a global scale to diagnose COVID-19 with utmost accuracy. Targeted therapies largely focus on interfering with or reducing the binding of viral receptors and host cell receptors.

Today, the most common pharmacological interventions for COVID-19 include antivirals, steroid treatment, and monoclonal antibodies. By inhibiting viral RNA synthesis, Remdesivir helps to combat viral replication; however, conflicting studies on the efficacy, side effect profile, and drug interactions prove that this drug requires more research. Steroids such as dexamethasone are important, primarily in patients requiring respiratory support, due to their anti-inflammatory effects on the lung pleura. However, dexamethasone can cause many severe side effects and also suppresses T cells and B cells, resulting in a decreased immune response to the virus itself and delayed production of antibodies. Monoclonal antibodies have been shown to decrease viral entry and replication, although usually requiring multiple different monoclonal antibodies to significantly decrease viral load. However, constructing an efficacious cocktail of monoclonal antibodies is not only difficult, but often very expensive and not widely available.

Overall, there is still a slew of knowledge left to be discovered about how we respond to the COVID-19 pandemic as a medical community, and there are new additions to our arsenal of knowledge every day. We are expected to respond with the latest in technology and research, and further studies into the pathogenesis, clinical implications, identification, diagnosis, and treatment of COVID-19 will help propel society past this pandemic.

**Author Contributions:** D.T.: "Introduction" & "Clinical Case Scenarios", S.D. (Sujay Dayal): "Treatment and Edits", P.P.: Pathogenesis, S.D. (Surbi Dayal): "Treatment and Edits", C.T.: "Proofreader and Editor", K.S.: "Proofreader and Editor", K.R.: "Vaccine and Diagnosis", G.C.: "Editing and additional research contributions to Introduction, Diagnosis, Vaccine, and Treatment Sections", A.M.: "Conclusion, abstract, and edits", R.S.: "Conclusion, abstract, and edits". V.V. worked with the team to create, draft, edit and proofread the manuscript. All authors have read and agreed to the published version of the manuscript.

**Funding:** We appreciate the funding support from National Institutes of Health (NIH) award RHL143545-01A1.

**Informed Consent Statement:** Informed consent was obtained from all subjects involved in the study.

**Data Availability Statement:** Data sharing not applicable. No new data were created or analyzed in this study. Data sharing is not applicable to this article.

**Conflicts of Interest:** The authors declare no conflict of interest.

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
