# Peer review of "Analysis of COVID-19 on Diagnosis, Vaccine, Treatment, and Pathogenesis with Clinical Scenarios"

_clinpract, doi:10.3390/clinpract11020044_

Round 1

Reviewer 1 Report

1: line 58-59

The recent worldwide pandemic, COVID - 19, 58 has intensely emphasized the key role of diagnostic technologies in "the containment" of infectious diseases

Q:This sentence is vague, and needs further explanation to make it clear.

2: line 74-75

Since access to portable machines are limited, CT scans are commonly utilized.

Q: access to portable machines are limited? what do you mean? You mean the portable machines are less than CT scans? or others?

Personally, what I know is that the information provided by the CT scans is more and fruitful than the portable X ray. I am not sure this is what you tried to explain?

3:line 88-95

RT-PCR: only for the detection of the RNA, and the results are positive or negative.

qPCR, quantitative polymerase chain reaction, or real-time PCR, (that is your so call "rRT-PCR) not only can identify the target, but also measure the amount of the RNA (viral load). This method can provide excessive information comparing to the RT-PCR.

4: 3 diagnosis

authors should provide data regarding the differences of positive rates between oral swab and nasal swab and explain the differences and recommend which one is better.

5: 4. Vaccine

Based on the information from the WHO website (https://www.gavi.org/covid19-vaccines?gclid=EAIaIQobChMIw-2InNPZ7wIVWX8rCh0GeQOSEAAYAiAAEgLrK_D_BwE)

There are four types of vaccines in clinical trials: whole virus, protein subunit, viral vector and nucleic acid (RNA and DNA), each of which protects people, but by producing immunity in a slightly different way.

COVID-19 vaccine types in development

Candidates in Clinical Phases I-III

Whole virus

15

Protein subunit

13

Nucleic

20

Viral vector

15

There are at least 63 types of vaccinations.  It is too shallow for authors to introduce 2 types of them.

  1. Treatment: how about an illustration of these medications? please add the role of IL-10 and TGF-b in this illustration for your context in the Clinical Case Scenario

Author Response

Responses to the comments of the reviewers:

We would like to first off thank you for taking your time off to read our paper “Analysis of COVID-19 on diagnosis, vaccine, treatment, and pathogenesis with clinical scenarios” especially during these COVID times. Before elaborating on the edits made on the paper, we would like to tell you a little bit about ourselves. We are 10 students: 6 medical students from Western University of Health Sciences Pomona, 2 undergraduate students from California State University Northridge, 1 undergraduate student from University of California Irvine, and 1 undergraduate student from Pitzer College under the guidance of Dr. Venketaraman. Dr. Venketaraman does extensive research on infectious diseases such as tuberculosis, and he encouraged us to do research and explore the topic of COVID-19 as this novel disease needs better elucidation and a comprehensive analysis for the scientific community and laypeople to comprehend. We hope we can find our manuscript for consideration in the journal Clinics and Practice, and we thank you again for providing excellent suggestions to enhance the quality of our manuscript. We have made the appropriate changes that are denoted in track changes. Please see the responses below for the individual comments provided.

Reviewer 1:

  1. line 58-59 “The recent worldwide pandemic, COVID - 19, 58 has intensely emphasized the key role of diagnostic technologies in the containment of infectious diseases” This sentence is vague and needs further explanation to make it clear.

We have addressed the sentence and expanded on the key technologies used for diagnosing COVID19

2: line 74-75 “Since access to portable machines are limited, CT scans are commonly utilized” Access to portable machines are limited? What do you mean? You mean the portable machines are less than CT scans? Or others? Personally, what I know is that the information provided by the CT scans is more and fruitful than the portable X ray. I am not sure this is what you tried to explain?

We have expanded on the use of CT scans and have clarified this sentence.

3: line 88-95 RT-PCR: only for the detection of the RNA, and the results are positive or negative. qPCR, quantitative polymerase chain reaction, or real-time PCR, (that is your so call "rRT-PCR”) not only can identify the target, but also measure the amount of the RNA (viral load). This method can provide excessive information comparing to the RT-PCR.

  We have modified this line and have clarified the use of RT-PCR

  1. diagnosis authors should provide data regarding the differences of positive rates between oral swab and nasal swab and explain the differences and recommend which one is better.

 In the diagnosis section, we added studies that described the accuracy of nasopharyngeal swab tests vs. saliva swab test with studies validating the accuracy of the saliva test over the nasopharyngeal test. Recommending nasopharyngeal as the more accurate method.

  1. Vaccine Based on the information from the WHO website (https://www.gavi.org/covid19-vaccines?gclid=EAIaIQobChMIw-2InNPZ7wIVWX8rCh0GeQOSEAAYAiAAEgLrK_D_BwE)

There are four types of vaccines in clinical trials: whole virus, protein subunit, viral vector and nucleic acid (RNA and DNA), each of which protects people, but by producing immunity in a slightly different way.

 In the vaccines section we described and discussed four other types of vaccines in addition to Pfizer and Moderna. An example of each of the four vaccines as well as its current phase in testing and efficacy established from Phase I tests.

Reviewer 2 Report

The authors submitted a clinical review on various aspect of COVID-19. The manuscript is well-written,. I have the following comments:

1) ABSTRACT: Vaccines should be briefly commented In the abstract

2) ABSTRACT: Tocilizumab should be cited among the fundamental drugs against COVIDD-19 (other comments on this below)

3) Line 55 pag. 3: Why the authors decided this approach? Using only 25 articles? this can be prone to bias related to selection of sources. More information about these sources should be provided, or this sentence should be deleted

4) DIAGNOSIS: The authors may mention salivar tests that may be important for the future

5) Among the instrumental tool for the diagnosis, echo lung may be briefly mentioned

6) I suggest to move down the paragraph about instrumental diagnosis after the lab tests

7) Vaccines apart from Pfizer and Moderna should be mentioned

8) Regarding treatment, Recommendation from updated Guidelines and living systematic reviews should be reported and discussed (e.g. https://doi.org/10.1136/bmj.n858)

9) Results from RECOVERY and REMAP trials should be mentioned. The authors should discuss tocilizumab as new important data confirmed its importance

10) I suggest moving the pathophysiological paragraphs before the treatment

11) Line 357 pag. 9: Disagree. Data confirm the importance of tocilizumab 10.1056/NEJMoa2100433

12) Diagnosis of COVID-19 is also based on clinical signs and symptoms. I may suggest to briefly described clinical phenotypes as depicted by the WHO

Author Response

We would like to first off thank you for taking your time off to read our paper “Analysis of COVID-19 on diagnosis, vaccine, treatment, and pathogenesis with clinical scenarios” especially during these COVID times. Before elaborating on the edits made on the paper, we would like to tell you a little bit about ourselves. We are 10 students: 6 medical students from Western University of Health Sciences Pomona, 2 undergraduate students from California State University Northridge, 1 undergraduate student from University of California Irvine, and 1 undergraduate student from Pitzer College under the guidance of Dr. Venketaraman. Dr. Venketaraman does extensive research on infectious diseases such as tuberculosis, and he encouraged us to do research and explore the topic of COVID-19 as this novel disease needs better elucidation and a comprehensive analysis for the scientific community and laypeople to comprehend. We hope we can find our manuscript for consideration in the journal Clinics and Practice, and we thank you again for providing excellent suggestions to enhance the quality of our manuscript. We have made the appropriate changes that are denoted in track changes. Please see the responses below for the individual comments provided.

Reviewer 2:

Hello. We would like to first off thank you for taking your time off to read our paper “Analysis of COVID-19 on diagnosis, vaccine, treatment, and pathogenesis with clinical scenarios” especially during these COVID times. Before elaborating on the edits made on the paper, we would like to tell you a little bit about ourselves. We are 10 students: 6 medical students from Western University of Health Sciences Pomona, 2 undergraduate students from California State University Northridge, 1 undergraduate student from University of California Irvine, and 1 undergraduate student from Pitzer College under the guidance of Dr. Venketaraman. Dr. Venketaraman does extensive research on infectious diseases such as tuberculosis, and he encouraged us to do research and explore the topic of COVID-19 as this novel disease needs better elucidation and a comprehensive analysis for the scientific community and laypeople to comprehend. We hope we can find our manuscript for consideration in the journal Clinics and Practice, and we thank you again for providing excellent suggestions to enhance the quality of our manuscript. We have made the appropriate changes that are denoted in track changes. Please see the responses below for the individual comments provided.

1) ABSTRACT: Vaccines should be briefly commented In the abstract

Vaccines are now discussed in abstract.

2) ABSTRACT: Tocilizumab should be cited among the fundamental drugs against COVIDD-19 (other comments on this below)

 Tocilizumab was discussed. See comments below

3) Line 55 pag. 3: Why the authors decided this approach? Using only 25 articles? this can be prone to bias related to selection of sources. More information about these sources should be provided, or this sentence should be deleted

We have used a total of 43 articles for the paper by different authors and researchers decreasing potential for selection bias and increasing internal validity.

4) DIAGNOSIS: The authors may mention salivar tests that may be important for the future

 Saliva swab tests were discussed as more accurate than nasopharyngeal swab test based on several studies.

5) Among the instrumental tool for the diagnosis, echo lung may be briefly mentioned

After extensive research, we could not find comprehensive research on echo lung utilization in COVID 19 Diagnosis.

6) I suggest to move down the paragraph about instrumental diagnosis after the lab tests

 We moved down the paragraph about instrumental diagnosis after the lab tests.

7) Vaccines apart from Pfizer and Moderna should be mentioned

 Four additional vaccines were discussed including examples and current efficacy based on undergone phase.

8) Regarding treatment, Recommendation from updated Guidelines and living systematic reviews should be reported and discussed (e.g. https://doi.org/10.1136/bmj.n858)

 We have the treatment recommendations updated using COVID guidelines.

9) Results from RECOVERY and REMAP trials should be mentioned. The authors should discuss tocilizumab as new important data confirmed its importance

 Tocilizumab discussed with efficacy of both RECOVERY and REMAP with data

10) I suggest moving the pathophysiological paragraphs before the treatment

 We moved the pathophysiological paragraph before the treatment.

11) Line 357 pag. 9: Disagree. Data confirm the importance of tocilizumab 10.1056/NEJMoa2100433

 Data discussed for Tocilizumab from RECOVERY and REMAP studies

12) Diagnosis of COVID-19 is also based on clinical signs and symptoms. I may suggest to briefly described clinical phenotypes as depicted by the WHO

 Signs and symptoms are discussed directly from CDC website.

Round 2

Reviewer 1 Report

Author Contributions: should be addeded

Institutional Review Board Statement and Informed Consent Statement:: can be waived

Data Availability Statement: I suggest the authors should put on some words to explain how they acquire their data

Acknowledgments: The authors did not have any comments on this part.

Reviewer 2 Report

No further queries